

# Bias correction in assimilation of AOD observations with WRF-Chem

Anton Kliewer[1], Milija Zupanski[1], Qijing Bian[2], Sam Atwood[2], Yi Wang[3,4,5], and Jun Wang[3,4,5]

[1]Cooperative Institute for Research in the Atmosphere, Colorado State University, Fort Collins, CO, USA
[2]Department of Atmospheric Science, Colorado State University, Fort Collins, Colorado USA
[3]Department of Chemical and Biochemical Engineering, The University of Iowa, Iowa City, IA USA
[4]Center of Global and Regional Environmental Research, The University of Iowa, Iowa City, IA USA
[5]Interdisciplinary Graduate Program in Informatics, The University of Iowa, Iowa City, IA USA

**Correspondence:** Anton Kliewer (anton.kliewer@colostate.edu)

**Abstract.** Accurate prediction and representation of three-dimensional aerosol distributions in the littoral (coastal) zone is both desired and difficult with many compounding factors contributing to this problem. To reduce uncertainty in forecasting in coastal regions, a coupled meteorological-aerosol data assimimilation (DA) system has been configured to include satellite observations of aerosol optical depth (AOD). These high-resolution observations are from newly-devised retrieval algorithms that utilize Moderate Resolution Imaging Spectroradiometer (MODIS) data to retrieve AOD over the coastal and turbid water surface. The Weather Research and Forecasting model coupled with Chemistry (WRF-Chem) is combined with an ensemble-based DA system, the Maximum Likelihood Ensemble Filter (MLEF), to simulate a dust event over the Arabian Peninsula from 2016. The assimimilation of AOD observations required the development of a forward operator that converts model predictions into observation space. This operator, which incorporates hygroscopic growth of aerosol particles and determines extinction efficiency based via Mie theory, has a positive bias between the model guess and the retrieved AOD observations. In order to reduce this bias two different methods are proposed and evaluated. One is a "moving average" method employed throughout the case study while the other relies on a statistical re-sampling approach. The conclusion of these experiments, determined by a number of metrics including, but not limited to, root mean square (RMS) errors, an evaluation of the reduction in the cost function, and degrees of freedom for signal (DFS), indicate that the bias reduction scheme that accumulates bias information throughout the case study outperforms the method based on re-sampling. This conclusion is corroborated by inspection of the analysis increments from the DA process and by the innovations in observational space. An analysis of the non-Gaussian innovations resulting from the non-linear forward operator is also presented. This research is in support of a Multidisciplinary University Research Initiative (MURI) supported by the Office of Naval Research (ONR) with the primary goal of understanding aerosols in the littoral zone.



# 1 Introduction

In 2015 a Multidisciplinary University Research Initiative (MURI) award was granted to a team of university researchers with the goal of formulating a better understanding, representation, and prediction of three-dimensional (3D) aerosol distributions in the littoral zone. The team's makeup represents a "holistic approach," in that researchers from both the observational and
modeling community will work together alongside data assimilation (DA) experts in order to attack the problem from many angles. This research is focused upon on how DA can provide high-resolution 3D analyses of aerosols in the littoral zone.

The goal of data assimilation is to marry model predictions with observational data to create an *analysis*, or true state, of the atmosphere. Therefore, to aid the overall understanding of aerosol distributions in the littoral zone, Moderate Resolution Imaging Spectroradiometer (MODIS) aerosol optical depth (AOD) observations are directly assimilated into a coupled
meteorological-aerosol ensemble-based data assimilation system.

Upon initial testing of the coupled DA system, it was apparent that a *bias* exists between the observational data and the model guess. Bias refers to the difference between the true value and the one predicted by an estimator. In DA this estimator converts the model variables into an equivelent form of the observational data. This is done by the *forward operator* which was developed and implemented as part of the MURI. Biases are inherit in any DA system (Aulignè et al., 2007) and arise from the
satellite instrument, the radiative transfer model describing the state of the atmosphere, and systematic errors inherant in the background state produced by a numerical weather prediction model. Aulignè et al. (2007) also contains a brief overview of the origins of bias correction relative to data assimilation.

Biases in MODIS AOD retrievals, compared against the ground-based Aerosol Robotic NeTwork (AERONET), have previously been described (Chu et al., 2002; Remer et al., 2005; Levy et al., 2013; Drury, 2008; Wang et al., 2010; Hyer et al., 2011;
Lary et al., 2009; Albayrak et al., 2013). Also, a study of biases in the littoral zone has also been characterized (Anderson et al., 2013). How these biases could affect data assimilation has also been considered (Zhang and Reid, 2006; Hyer et al., 2011; Shi et al., 2013) and, as noted in Reid et al. (2009), these AOD biases can have a large impact on the analysis produced from DA. While much of the previous research has been devoted to improving the quality of the AOD retrievals, to the authors' knowledge no DA system has included a bias correction scheme.

To account for the bias with respect to AOD observations, two online methods are incorporated into the DA system and tested on a dust event over the Arabian Peninsula in 2016. This case study includes two large dust plumes originating over a 36-hour period. These plumes are generated by different dynamic forcing mechanisms and are located in and around the littoral zone, and thereby formulate an ideal test case to suit the goals of the MURI. To fully understand the AOD bias inherit within the DA system in re-creating this case study, the controls variables are the relevant aerosol species particular to the chemical
aerosol module used in the model simulation.

This specific research regarding the assimilation of aerosol observations is a contribution to the Holistic Analysis of Aerosols in Littoral Environments Multidisciplinary University Research Initiative (HAALE-MURI) Atmospheric Chemistry and Physics/ Atmospheric Measurement Techniques (ACP/AMT) Special Issue. The remainder of this paper is organized as follows: section 2 contains details regarding methodology, section 3 contains results, and section 4 includes conclusions and a summary.



## 2 Methodology

### 2.1 Model

The Weather Research and Forecasting (WRF) model version 3.9.1, a mesoscale numerical weather prediction system coupled with chemistry (WRF-Chem) (Grell et al., 2005; Fast et al., 2006), is used to simulate a 2016 dust event over the Arabian

Peninsula. Details regarding this dust event are discussed in section 2.5. The region of interest is shown in Fig. 1, with the outer and inner domains having a resolution of 27 km and 9 km respectively. The outer domain spans approximately 4600 km x 3700 km while the inner domain covers a 1500 km x 1500 km area. There are 50 vertical levels with a model top of 10 hPa. Within WRF the Goddard microphysics (Tao et al., 1989) and Kain-Fritsch (Kain, 2004) convective parameterization schemes are selected. The MM5 similarity scheme (Jiménez et al., 2012) is chosen for the surface layer processes and the Noah Land

Surface model (Tewari et al., 2004) for surface processes. Meteorological lateral boundary conditions (LBC) were provided by the National Centers for Environmental Prediction (NCEP) Global Forecast System (GFS).

The Goddard Global Ozone Chemistry Aerosol Radiation and Transport (GOCART) aerosol module (Chin et al., 2000, 2002; Ginoux et al., 2001) is utilized for aerosol processes. With GOCART, forecasts of 3D mass concentrations are provided for fourteen aerosol species: hydrophobic and hyrophilic black carbon (BC), hydrophobic and hyrophilic organic carbon (OC),

sulfate, sea salt in four size bins (effective radii of 0.3, 1.0, 3.2, and 7.5 $\mu$m for dry air), and dust in five size bins (effective radii of 0.5, 1.4, 2.4, 4.5, and 8.0 $\mu$m for dry air). The GOCART scheme also provides dust (Ginoux et al., 2001) and sea salt emissions.

### 2.2 Data Assimilation

The Maximum Likelihood Ensemble Filter (MLEF; Zupanski, 2005; Zupanski et al., 2008) is used to assimilate AOD obser-

vations. The MLEF finds the maximum of the posterior probability density function by minimizing the Gaussian-based cost function given by:

$$J = \frac{1}{2}(\boldsymbol{x} - \boldsymbol{x}_f)^\mathsf{T}\mathbf{P}_f^{-1}(\boldsymbol{x} - \boldsymbol{x}_f) + \frac{1}{2}(\boldsymbol{y} - h(\boldsymbol{x}))^\mathsf{T}\mathbf{R}^{-1}(\boldsymbol{y} - h(\boldsymbol{x})) \tag{1}$$

where the subscript $f$ refers to the forecast and **x** and **y** define the state and observation vectors respectively. The diagonal matrix **R** is the observation error covariance. The transpose of a matrix is represented by T. The non-linear observation

operator, or forward operator, $h$ is discussed in section 2.3.2. The $i$-th column of the square root *forecast* error covariance is defined as

$$\mathbf{p}_f^i = m\left(\boldsymbol{x}_a + \boldsymbol{p}_a^i\right) - m\left(\boldsymbol{x}_a\right) \tag{2}$$

where the subscript $a$ refers to the analysis, $\mathbf{p}_a^i$ is the $i$-th column of the square root analysis error covariance and $m$ is the nonlinear forecast model.





The control variables includes east-west and north-south wind components, perturbation potential temperature, water vapor mixing ratio, perturbation dry air mass in column, cloud water mixing ratio, rain water mixing ratio, ice mixing ratio, snow mixing ratio, the five GOCART dust species, and the four GOCART sea salt species. The MLEF is coupled in the sense that meteorological control variables can affect AOD increments and vice versa. A flow chart for the MLEF is seen in Fig. 2 and for more details regarding the WRF-Chem coupled DA system see Zupanski (2018).

As there appears to be a systemic difference between $\boldsymbol{y}$ and $h(\boldsymbol{x})$, which we classify as a bias $b$, the two proposed methods will attempt to account for this by adjusting the model guess $h(\boldsymbol{x})$. Specifically, the observation model given by

$$\boldsymbol{y} = h(\boldsymbol{x}) + \boldsymbol{\eta}, \tag{3}$$

where $\boldsymbol{\eta}$ is a Gaussian random variable with zero mean and variance given by $\mathbf{R}$, should include the bias term and the forward operator would therefore be

$$\tilde{h}(\boldsymbol{x}) = h(\boldsymbol{x}) + \boldsymbol{\eta} + b \cdot \mathbb{1} \tag{4}$$

where $\mathbb{1}$ is a vector of 1's. Hence, the cost function (1) would therefore be

$$J = \frac{1}{2}(\boldsymbol{x} - \boldsymbol{x}_f)^{\mathsf{T}} \mathbf{P}_f^{-1}(\boldsymbol{x} - \boldsymbol{x}_f) + \frac{1}{2}(\boldsymbol{y} - h(\boldsymbol{x}) - b)^{\mathsf{T}} \mathbf{R}^{-1}(\boldsymbol{y} - h(\boldsymbol{x}) - b). \tag{5}$$

As a result, ideally the "analysis increments," defined by $\boldsymbol{x}_a - \boldsymbol{x}_b$, which is the difference between the analysis and background state vectors, would follow a symmetric, Gaussian distribution centered around 0.

## 2.3 Observations

### 2.3.1 AOD Data

AOD observations from the Moderate Resolution Imaging Spectroradiometer (MODIS) collection 6.1 (https://modis-atmos. gsfc.nasa.gov/sites/default/files/ModAtmo/C061_Aerosol_Dark_Target_v2.pdf) are used in our experiments. MODIS 3-km resolution AOD products over land and ocean are derived by Dark Target Land (DT-land) (Munchak et al., 2013; Remer et al., 2013) and Dark Target Ocean (DT-ocean) algorithms (Remer et al., 2013) respectively. Specifically, 3-km AOD retrievals at 0.55 $\mu$m are used in the data assimilation step.

In addition to the DT-land and DT-ocean retrievals, additional AOD observations over turbid coastal water regions are also assimilated. In the collection 6.1 product, AOD retrievals over turbid coastal water are not available due to high water leaving radiance at 0.55 $\mu$m, 0.66 $\mu$m, and 0.86 $\mu$m (Li et al., 2003). Taking advantage of the fact that water leaving radiance at 2.1 $\mu$m is negligible, Wang et al. (2017) introduced an algorithm of using MODIS Top of Atmosphere (TOA) reflectance at this band to retrieve AOD over turbid coastal water. Not only are AOD retrieval gaps over turbid coastal water are filled, but it





also shows a combination of AOD retrievals from this algorithm and dark target algorithms can improve data quality when validating against AERONET measurements (Wang et al., 2017). These AOD observations are also retrieved at the 0.55 $\mu$m band with 3-km resolution.

Since AOD observations are only available in daytime, Fig. 3 describes the two DA assimation steps per day where this
observational data is used in our case study, from MODIS/Terra and MODIA/Aqua respectively, at local 11:30 a.m. and 1:30 p.m.

### 2.3.2 Forward Operator

The AOD forward operator $h$ computes an AOD value $\tau$ from the WRF-Chem state $\boldsymbol{x}$ to compare with the observational values $\boldsymbol{y}$. These values are then used in the assimilation step through the minimization of Eq. 1. Similar to the observation operator
described in detail in (Wang et al., 2004; Liu et al., 2011), we assume all of GOCART aerosol species (dust, sea salt, organic carbon, black carbon, sulfate) are spherical, externally mixed and that their size distributions are lognormally distributed. To compute a mass extinction coefficient $E_{ext}$ (with units $m^2 g^{-1}$) for each particular species, the refractive index $n_{r_i}$ and effective radius $r_{eff_i}$ must be defined. Table 1 contains these assumptions along with each species' size properties. Hygroscopic growth of hydrophilic aerosol species is accounted for using $\kappa$-Köhler theory (Petters and Kreidenweis, 2007) to grow particles to
equilibrium with modeled relative humidity values. After accounting for hygroscopic growth, the wavelength- ($\lambda$) dependent $E_{ext}$ is computed using Mie theory. To compute total column AOD, we sum over all model layers $k$ and $n = 14$ GOCART aerosol species with the following equation given in Pagowski et al. (2014):

$$\tau(\lambda) = \sum_{i=1}^{n} \sum_{k=1}^{ktop} E_{ext}(\lambda, n_r, r_{eff_i}) \times c_{ik} \times \rho_{d_k} \times d_k \tag{6}$$

where $c$ is aerosol mixing ratio ($\mu$g/kg of dryair), $\rho_d$ is the dry air density, and $d$ is the layer depth.
Table 1 also contains the specified hygroscopic growth factors $\kappa$. The observational errors used within the diagonal matrix $\mathbf{R}$ are defined to be 0.300.

### 2.4 Bias Correction

Preliminary experiments found that AOD innovations ($\boldsymbol{y} - h(\boldsymbol{x})$), which is the difference between the observations and model guess, had a strong, positive bias. This can be seen in Fig. 4 with AOD observations in the littoral zone being larger than the
model guess. Figure 4 also contains a scatter plot of the model guess against the observations. From this it is evident that there is a positive bias between these two quanitites as there is a large number of points falling above the line shown in red. In an attempt to account for this bias, two different types of bias correction scheme are employed for this case study. The following two subsections detail these schemes and section 3 describes their performance.





### 2.4.1 Moving Average

The first bias correction scheme is referred to as the "moving average." In this approach the bias is defined to be the mean of the innovations. This bias accumulates from cycle to cycle, where for each particular cycle, the bias is found from all of the innovations from the current and previous cycles. This method may be thought of as "situation-dependent," as noted in Dee

(2006). It is posited that this moving average (or cumulative bias) may be ideal for one case study in a particular region, i.e., that this bias may encapsulate the model and reprensentative error of the AOD assimilation in the littoral zone of the Persian Gulf during this dust event.

Mathematically, the moving average approach includes a recursive bias update formula:

$$b_{k+1} = \alpha \cdot b_k + (1 - \alpha) \cdot b_{0,k-1} \qquad (7)$$

where $b$ denotes bias, $\alpha$ is a recursive coefficient, and index $k$ denotes the DA cycle. The value $b_{0,k-1}$ essentially represents the bias from all previous cycles, while $b_k$ is the new bias obtained using AOD observations from the current cycle. For any $k = 0, 1, \ldots$ the current bias is $b_k = \frac{1}{N_k} \cdot \sum (\boldsymbol{y} - h(\boldsymbol{x}))_k$, where $N_k$ is the number of AOD observations in the $k$-th cycle. This bias $b_k$ is then added to the model guess prior to the assimilation step, as noted in the equation (5).

### 2.4.2 Bootstrap

Since only a small number of observations may be available in the inner domain after quality control, a bootstrap bias correction scheme attempts to exploit as much information from this sample as possible. With the set of innovations treated as the population, a sample is taken (with replacement) from the innovations. Each sample is of equal size to the set of innovations. This is repeated $M$ times. The mean of the $i^{th}$ sample, denoted by $\bar{z}_i$, is calculated, and then the mean of these means is found. This value is defined to be the bias, i.e. $b = \frac{1}{M} \cdot \sum_{i=1}^{M} \bar{z}_i$. Each model guess is then adjusted with this bias similar to the moving

average scheme. Experiments were run with $M = 10,000$ and $M = 100,000$ to evaluate how dependent the analyses are on this parameter.

## 2.5 Experiments

To evaluate the efficacy of the bias corrrection schemes, a dust storm over the Arabian Peninsula during 3-4 August 2016 is chosen as our case study. In this event two large dust plumes are generated by different dynamic forcing mechanisms and reside

in very different types of air masses. As seen in Fig. 5, the MODIS true color image shows one plume over the Persian gulf and another over central Saudi Arabia. This case study is a useful test of the bias correction methods' impact on the model analyses representation of the fundamental processes that govern aerosol distribution in a littoral zone. More details of this dust event can be found in Miller et al. (2018).

To encapsulate the formation and evolution of these dust plumes, the WRF-Chem simulation begins at 0000 UTC 3 August

2016 and as previously mentioned observations are assimilated twice daily during the event as depicted in Fig. 3. In these experiments the MLEF uses 32 ensemble members.





Each bias correction scheme is employed through an entire WRF-Chem simulation and data assimilation experiment. The test results from the different schemes will be labeled as follows: the moving average *(MA)*, the bootstrap scheme with $M = 10,000$ *(BS)*, and the bootstrap scheme with $M = 100,000$ *(BL)*. As a control, these schemes will be compared to an experiment without any form of bias correction *(NO)*.

## 2.6 Diagnostics

Several quantitative metrics will be analyzed comparing the results of the experiments with and without bias correction and are defined below.

*Number of observations ($N_{obs}$).* This value indicates the number of observations that were assimilated in each DA cycle.

*Cost function ($\%\Delta_{cost}$).* The cost function (1) is evaluated with AOD observations with the background and analysis state and the relative change is found (as a percentage). Ideally, there should be a marked decrease in this value.

*Root mean square (RMS) error ($\%\Delta_{RMS}$).* RMS errors for the analysis and background, denoted with the subscripts $a$ and $b$ respectively, are defined by the following equations:

$$RMS_a = \sqrt{\frac{1}{n}\sum_{i=1}^{N_k}(\boldsymbol{y}-h(\boldsymbol{x}_a)^2}$$

$$RMS_b = \sqrt{\frac{1}{n}\sum_{i=1}^{N_k}(\boldsymbol{y}-h(\boldsymbol{x}_b)^2}. \tag{8}$$

The relative change is then found (as a percentage) and once again a large, negative value is sought.

*Skill.* Skill is calculated as

$$skill = \frac{1}{N_{obs}\cdot N_{ens}}\sum_{j=1}^{N_{ens}}\sum_{i=1}^{N_{obs}}(\boldsymbol{y}_i-h_i(\boldsymbol{x}_j))^2 \tag{9}$$

where $N_{ens}$ is ensemble size. Skill measures the distance between analysis and observations and represents the accuracy of the MLEF analysis to the given observations.

*Spread.* Spread (ensemble spread) is calculated as

$$spread = \sqrt{\frac{1}{N_{obs}\cdot N_{ens}}\sum_{j=1}^{N_{ens}}\sum_{i=1}^{N_{obs}}(h_i(\boldsymbol{x}_j)-h_i(\boldsymbol{x}_c))^2} \tag{10}$$

where the subscript $c$ stands for control. Spread measures the deviation of ensemble estimates from the control estimates and represents the uncertainty of the MLEF analysis.

*Spread over skill.* Spread divided by skill tries to encapsulate the two previous quantities. The relationship between spread and skill is often used in ensemble forecast verification as an indication of how well an ensemble is performing. In general, an





ideal situation would be to have comparable skill and spread because ensemble spread should be a good indicator of possible forecast error (skill) distributions that reflects the true predictability of a flow. In that sense, spread over skill equal to 1 is sought.

$\chi^2$. The $\chi^2$ validation test accumulates from cycle to cycle and is fully explained in Zupanski (2005). It is defined to be

$$\chi^2 = \frac{1}{N_{obs}} \left[ \boldsymbol{y}_k - h(\boldsymbol{x}_k) \right]^\mathsf{T} (\mathbf{H}\mathbf{P_f}\mathbf{H}^\mathsf{T} + \mathbf{R})^{-1} \left[ \boldsymbol{y}_k - h(\boldsymbol{x}_k) \right] \tag{11}$$

where $\mathbf{H}$ is the jacobian of the observation (forward) operator, i.e.

$$\mathbf{H} = \left( \frac{\partial h}{\partial \boldsymbol{x}} \right)_{\boldsymbol{x}_b}. \tag{12}$$

$\chi^2$ evaluates the correctness of the innovation covariance matrix that employs a predefined observation error covariance $\mathbf{R}$, and the MLEF-computed forecast error covariance $\mathbf{P}_f$. Values close to 1 are expected if the observation operator $h$ is linear and the innovations follow a Gaussian probability distribution. Due to the observation operator $h$ including hygroscopic growth, it is certainly non-linear and therefore we do not expect $\chi^2 = 1$.

These diagnostics, along with a presentation a model- and observational-space impact of AOD DA, are presented in the next section.

## 3   Results

The effectiveness of the bias correction schemes will be analyzed quantitatively in both model and observational space. Since we are interested in high-resolution simulations, only results from the inner domain will be presented. First, as a qualitative measure of the experiment design and execution, Fig. 6 represents the analysis at approximately the same time as the true color image in Fig. 5. This image shows the integrated dust for the five GOCART dust species and it is clear that the dust plumes are well represented and are in agreement with the MODIS image. The larger of the plumes is clearly seen over the Persian Gulf.

Table 2 contains the biases found for each scheme per assimilation cycle with AOD observations. These values indicate how the model guess is adjusted prior to the the maximization of the cost function (Eq. 1). From this table it is evident that both the BL and BS schemes compute essentially the same bias. For this reason, and since multiple other diagnostics seem to indicate similar results between the BL and BS schemes, only results from the BL experiment will be presented. Note that in the last two assimilation cycles that the bias associated with MA experiment is less than half of that found for the BS and BL tests. Results of the diagnostics previously defined are found in Table 3 comparing the NO, MA, and BL experiments and are discussed below.

Notably MA allows for marginally more observations to be assimilated in cycles 6 and 7 while BL results in substantially less observations to pass quality control than NO. This is probably due to the magnitude of the bias found in these cycles, noted in Table 2. This phenomena will be discussed in more detail.





Regarding the change in the cost function ($\%\Delta_{cost}$) there does not appear to be a consistent trend within and between the experiments. NO performs the best earlier in the experiment but dramatically increases at the end, where MA outperforms the others. While the analysis from cycle 7 is of most import to the characterization of the dust plumes, results from all DA cycles are presented due to the generation and transport of the aerosol distributions thoughout the case study. While all tests indicate substantial reductions in RMS errors throughout the entirety of the experiments, no experiment clearly outperforms the others.

From inspecting skill it appears all experiments performed similarly except for BL in the last DA cycle and all experiments seem to be comparable for spread. Combining these two statistics in spread over skill BL performs the best in cycles 6 and 7 when the dust plumes are largely present. As expected values for $\chi^2$ are not equal to 1 and moreover the innovations also do not follow a Gaussian distribution as will be discussed in section 3.4.

The notable conclusions from these results include the reduced number of assimilated observations in the BL experiment and the relative change of $\%\Delta_{cost}$ for the MA experiment. Overall, since there does not appear to be a definitive conclusion as to which bias correction scheme is preferred, further analysis is required.

## 3.1  Information Theory

To quantify uncertainty reduction degrees of freedom for signal (DFS; Rodgers (2000)) is defined to be

$$d_s = tr\left[\mathbf{I} - \mathbf{P}_a\mathbf{P}_f^{-1}\right]$$

where $\mathbf{I}$ is the identity matrix and $tr$ is the trace of a matrix. DFS can also be expressed as

$$d_s = \sum_i \frac{\lambda_i^2}{(1 + \lambda_i^2)}$$

where $\lambda_i$ are the eigenvalues of the information matrix (Zupanski et al., 2007). DFS is a non-negative value with positive values indicating a reduction of uncertainty due to assimilating AOD observations. A value of $0$ indicates no impact of observations. Figure 7 shows the DFS at model level five for the NO, MA, and BL experiments and it appears that observations have the greatest impact in the experiment without bias correction. The largest values of $d_s$ are located in the Persian Gulf, coinciding with a majority of the observations. For MA and BL, there is also a reduction of uncertainty in the Persian Gulf, indicating a positive effect from assimilating AOD observations.

## 3.2  Model Space

Another way to assess the impact of assimilating AOD observations is to consider the analysis increments. This is the correction to the background state from the advent of observational data. Figure 8 depicts the analysis increments for the NO, MA, and BL experiments for the "Dust 2" species at model level $k = 6$. In the NO experiment it is clear that the large, positive values in the area of the dust plume over the Persian Gulf indicate that the model is under representing this dust event and that the AOD observations correct for this. The MA and BL bias correction schemes dampen the impact comparatively of these observations and in the case of the BL experiment, the increment is largely negative indicating that the model predicted more dust, for this particular species, than was observed.





This general trend is consistant for the other dust species and vertical levels. In consideration of how the increment changes vertically, Fig. 9 shows the average increment across the inner domain. Note that, similarly to Fig. 8, the NO experiment shows a stark positive increment, contrary to the BL experiment. The MA scheme appears to be the most centered around zero, possibly indicating that this spatiotemporal approach may be the most balanced. Figure 9 also shows that there is very little

bias correction above model level 22, indicating that the dust plumes are located within the troposphere.

### 3.3    Observation Space

In observation space the impact of the bias correction schemes can be viewed by considering the innovations, given by $(\boldsymbol{y} - h(\boldsymbol{x}))$, which represents the MODIS observation minus the model guess of AOD when an observation is assimilated. In Fig. 10, the innovations are shown for the NO, MA, and BL experiments in the last DA step (cycle 7).

Clearly, and as noted previously, there is a significant reduction in the number of assimilated observations in the BL experiment. It appears that the majority of these "lost" observations occurs over the Persian Gulf and in the littoral zone as opposed to over land. There are approximately 3 times as many AOD observations assimilated in the NO and MA experiments and cover more of the Persian Gulf. It is also seen that a majority of the innovations for the BL experiment are less than zero, possibly indicating that this scheme over-compensated for the bias and skewed the innovations negative. This is probably due to the

large bias value reported in Table 2.

Lastly, while the NO and MA experiments assimilate approximately the same number of observations, they do not cover same areas of the Persian Gulf, notably north of the island of Bahrain and the northern most part of the Gulf, east of Kuwait. These experiments do not reveal the cause of this phenomena and could require further study. However, when these experiments do assimilate observations in similar locations, they mostly agree on the sign and magnitude of the innovation.

### 3.4    Gaussian Assumption

The MLEF cost function Eq. 1 is based on the Gaussian distribution (Zupanski, 2005) and if the observation operator $h$ is linear then the innovations should follow a Gaussian distribution. However, $h$ is certainly non-linear since it incorporates hygroscopic growth as a function of relative humidity. As a result, in Fig. 11 it is clear that the innovations, in observation space for each DA cycle and bias correction scheme, typically follow a non-Gaussian probability distribution for the NO, MA,

and BL experiments.

For the NO experiment, shown in the top row of this figure, the positive bias is evident throughout the case study as the peak of the distribution is greater than zero. It was hoped that the bias correction schemes could reduce this non-Gaussian behavior and bias and produce more symmetric innovations. It appears that the MA scheme approaches these goals as the distribution in the last DA cycle appear more symmetric and centered around zero than in the NO experiment. Notably, in the last DA cycle,

the distribution of innovations for the BL experiment is heavily skewed and centered below zero. This validates the previous assertions that this scheme over-compensates for the bias.

To quantitatively compare these distrubitions to a Gaussian distribution the skewness and kurtosis, the third and fourth moments respectively, are found and described in Table 4. The Gaussian distribution has a skewness equal to 0 and kurtosis





equal to 3. In this table the values indicate that the NO experiment more closely resembles a Gaussian distribution most of the time. One clear conclusion is the excess skewness and kurtosis found in the last DA cycle for the BL scheme.

Similarly, histograms of the analysis increments, in model space, can be seen in Fig. 12. Here the trend continues that the NO experiment shows a positive skew and the BL a negative skew. The MA scheme shows the most symmetric distribution

centered around zero. Note that all increments equal to 0 have been removed to better represent the distribution of the correction to the background state.

While it was assumed that there would be a non-linear effect from the forward operator on the distribution of innovations, here it is evident how the different bias correction schemes impact this distribution throughout the case study. In this regard it may appear that the experiment without any bias correction or the moving average bias correction scheme may outperform the

bootstrap bias correction scheme.

## 4  Conclusions

In support of a MURI sponsored by the Office of Naval Research, two bias correction schemes are evaluated in assimilating AOD observations for a dust event over the Arabian Peninsula in August 2016. This research used WRF-Chem with GFS data for LBC to simulate the dust event and the MLEF to assimilate new, high-resolution MODIS AOD observations. The case

study analyzed included two large dust plumes, one over Saudi Arabia and the other over the Persian Gulf. This research is meant to aid the MURI goals of forecasting and understanding 3D aerosol distributions in the littoral regions.

The first bias correction scheme is described as a moving average that accumulates and incorporates the bias found in all previous assimilation cycles in an attempt to capture and correct for the regional and temporal representation error characteristics. The second bias correction scheme involves bootstrap sampling with the goal of determining the true statisics of a probability

distribution by exploiting a small sample of innovations. These two schemes were tested against an experiment that included no bias correction.

Overall, the three experiments represented the dust event quite well compared to a MODIS true color image. The simulation with no bias correction scheme revealed a positive bias between the AOD observations and the model guess. This bias was found to be present in all of the assimilation steps where AOD observations were available. Also, as a result of the non-

linear forward operator for the AOD observations, the distribution of innovations are non-Gaussian. The goal of the bias correction schemes was to account for and mitigate this phenomena. A variety of metrics evaluated their effectiveness including considering RMS errors, information theory, and distribution statistics.

The moving average bias correction method performed quite well in reducing the positive bias in the innovations. A variety of distributions, of both the innovations and the analysis increments, showed that the bias was more symmetric and closer to

zero than the NO experiment. This was found to be the case both vertically and across the two-dimensional domain. As a result the analysis increments included both positive and negative corrections to the background state. Using information theory, the DFS showed that the AOD observations were of substantial import in the littoral zone of the Persian Gulf. Also, the MA test



assimilated the most observations in the last two data assimilation cycles and resulted in the largest relative decrease in the cost function.

The bootstrap sampling scheme computed a bias that was over twice as large as the moving average scheme in the last two assimilation cycles. Therefore this method over-compensated for the bias between the observations and model guess. This

produced skewed distributions centered below zero. As a result, the observational impact from data assimilation was found to be lower than the NO and MA tests in the context of DFS. The analysis increments were predominately negative over the Persian Gulf indicating that the model precictions were dampened by the AOD observations. Another signature in the BL experiment was a reduced number of observations being assimilated over the course of the simulation.

As a result of these experiments it is concluded that the moving average bias scheme outperformed the bootstrap sampling

method in reducing the bias found between the observations and the model guess. The MA test also showed a minimal improvement over the experiment without bias correction. It is posited that further research may be desired in DA to account for the non-Gaussian behavior resulting from the non-linear forward operator. Lastly, these tests provide a novel understanding of aerosol distributions in the littoral zone after the assimilation of high-resolution satellite observations.

*Author contributions.*   AK and MZ contributed to implementing the forward model in the MLEF. QB and SA contributed in determining $E_{ext}$

for the GOCART aerosol species after experiencing hygroscopic growth. YW and JW provided AOD observations used in data assimilation. AK and MZ developed the bias correction schemes and AK conducted all of the experiments. A majority of the manuscript was written by AK with consultation from all other authors.

*Competing interests.*   The authors declare that they have no conflict of interest.

*Acknowledgements.*   This work is for a Department of Defense Multidisciplinary University Research Initiative and is supported by the Office

of Naval Research through contract N00014-16-1-2040 (Grant 11843919).



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

**Figure 1.** The outer and inner domain defined in WRF-Chem.





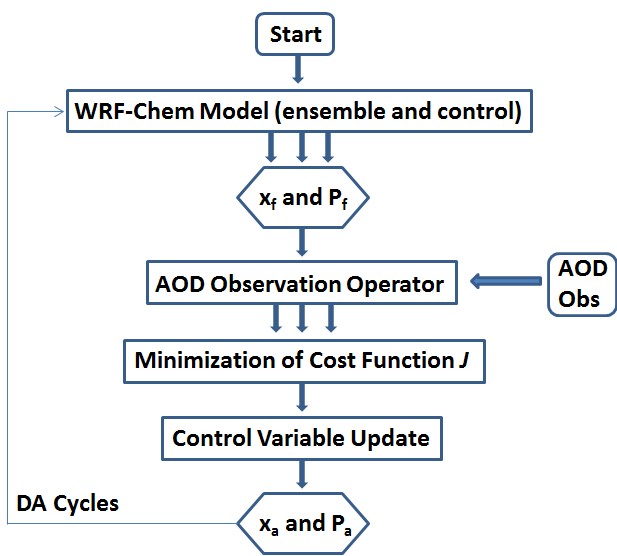

**Figure 2.** Flow chart for the MLEF DA system.





| Cycle | 1 | 2 | 3 | 4 | 5 | 6 | 7 |
|-------|---|---|---|---|---|---|---|
| Date | 0000 UTC 03 Aug 2016 | 0600 UTC 03 Aug 2016 | 1200 UTC 03 Aug 2016 | 1800 UTC 03 Aug 2016 | 0000 UTC 04 Aug 2016 | 0600 UTC 04 Aug 2016 | 1200 UTC 04 Aug 2016 |
| Obs | NO | YES | YES | NO | NO | YES | YES |

**Figure 3.** AOD observations are available and assimilated in daytime, corresponding to these DA cycles.




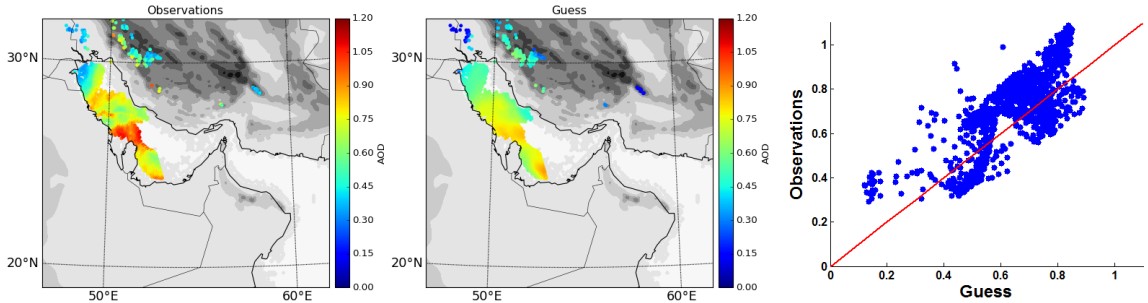

**Figure 4.** For the data assimilation step at 1200 UTC 4 August 2016 (cycle 7), the left image depicts the AOD observations and the middle image the model guess. On the right is a scatterplot between these two quantities, indicating a positive bias in the observations.




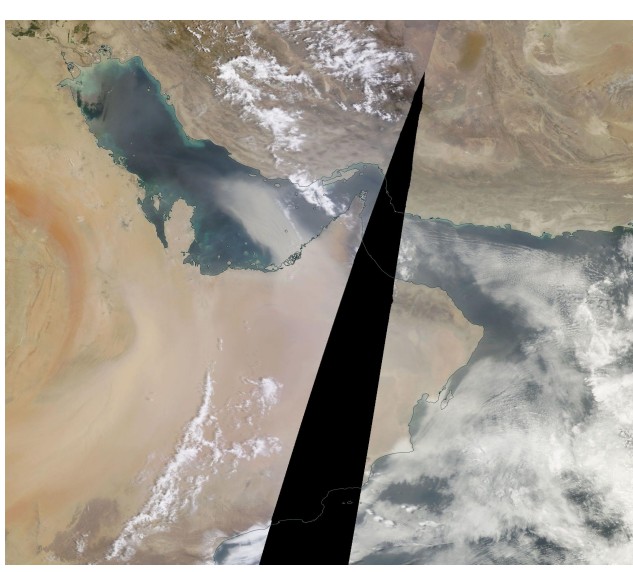

**Figure 5.** MODIS true color image of the two dust plumes over the Arabian Peninsula from 4 August 2016 at 1300 UTC. One plume extends across the United Arab Emirates into the Persian Gulf and was lofted from coastal Oman by a convective downburst. The second plume, in the center of the Arabian Peninsula, is embedded within a dry air mass.





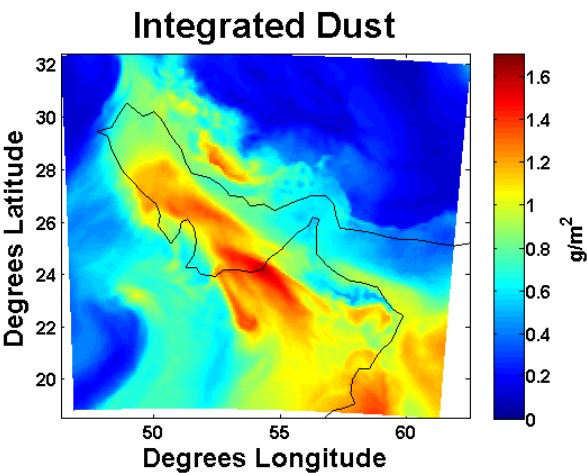

**Figure 6.** Integrated dust from the DA analysis at 1200 UTC 4 August 2016 without any bias correction scheme. Notice the two dust plumes in this inner domain, similar to the true color image in Fig. 5.




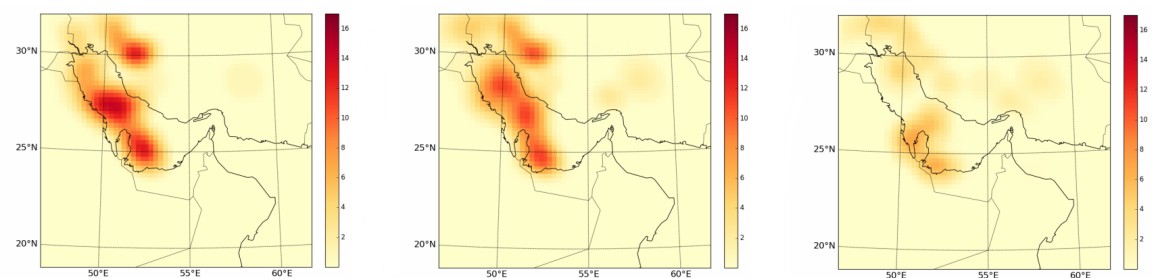

**Figure 7.** Degrees of freedom for signal, $d_s$, for the NO experiment (left), MA experiment (middle), and BL experiment (right). The units are non-dimensional.





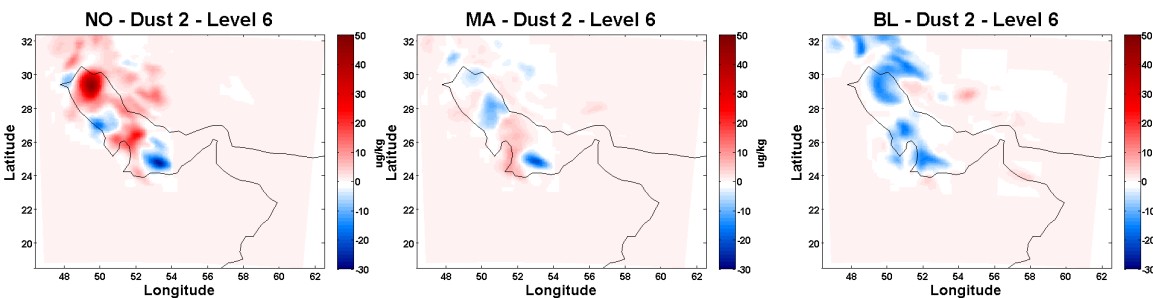

**Figure 8.** Analysis increments, $x_a - x_b$, in model space at 1200 UTC 4 August, 2016 for the NO, MA, and BL experiments. The NO experiment shows a large, positive correction in the dust plume over the Persian Gulf and a negative increment for the BL scheme. Notice that the MA scheme includes positive and negative increments for the Dust 2 species at model level 6 over the Gulf.





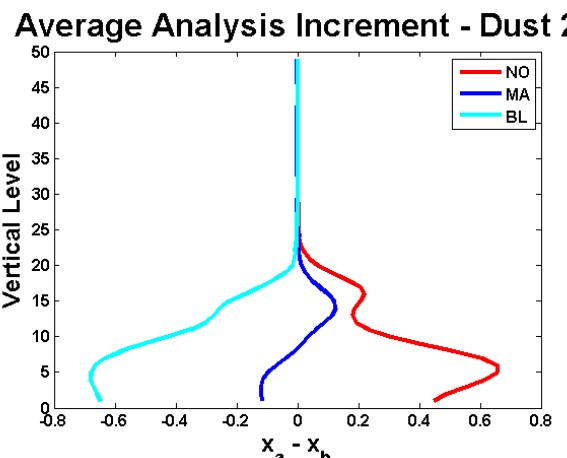

**Figure 9.** Average analysis increment over the inner domain for the Dust 2 species by vertical level. Notice the positive and negative values in the NO and BL schemes respectively and that the MA scheme is more centered around 0. The dust is found in the first 22 model levels and therefore there is no correction above this level.





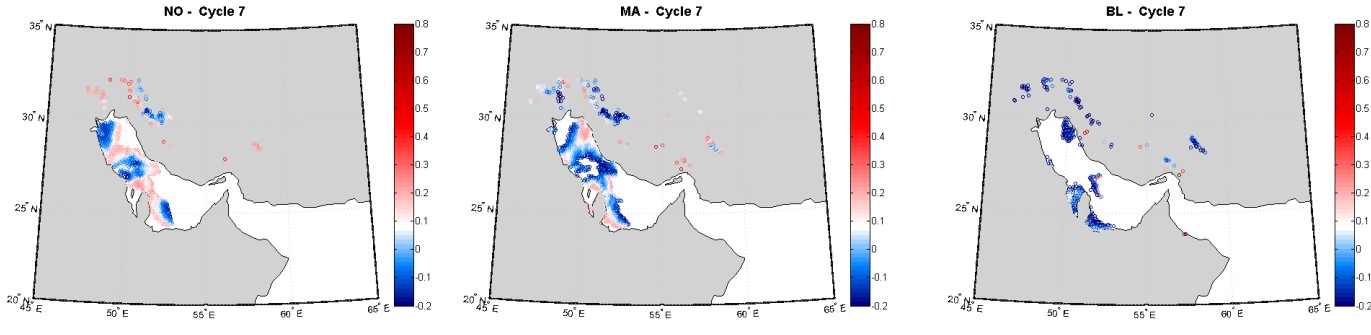

**Figure 10.** The innovations, $\boldsymbol{y} - h(\boldsymbol{x})$, at 1200 UTC 4 August, 2016 for the NO, MA, and BL experiments. This figure is similar to Fig. 8 except in observation space and shows similar postive and negative values for the NO and BL bias correction schemes over the Persian Gulf. The reduced number of assimilated observations is very apparent for BL here.





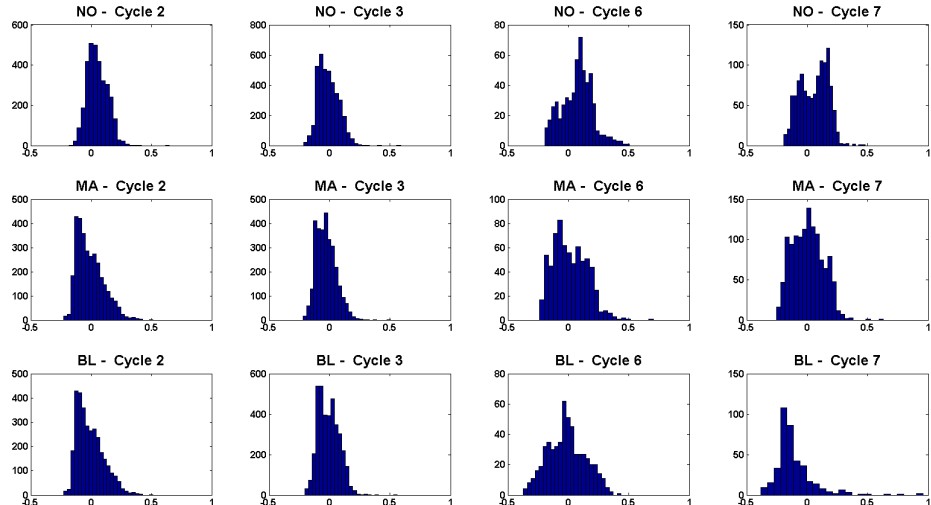

**Figure 11.** Histograms of the innovations, $y - h(x)$, in observation space, for the NO (top row), MA (middle row), and BL (bottom row) experiments for the four DA cycles where AOD observations are available. The NO experiment shows a positive bias throughout the four assimilation cycles whereas the MA scheme shows a more balanced distribution around zero. The BL scheme seems to over-compensate for the bias and shift the distribution to the left of zero in the last DA cycle. Also, due to the non-linear forward operator, these distributions are clearly not Gaussian.





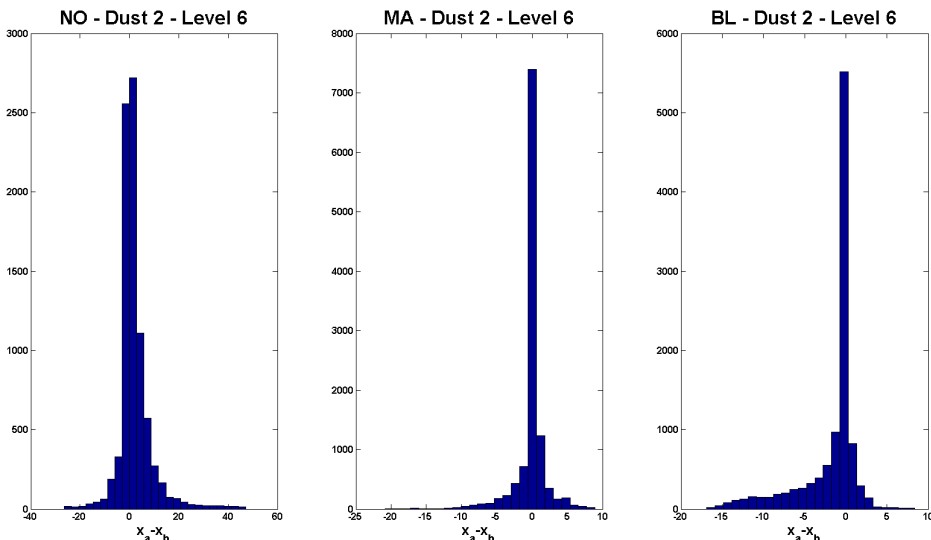

**Figure 12.** Histograms for the analysis increments shown in Fig. 8 clearly showing them to be non-Gaussian. The MA bias correction scheme seems to be more symmetric and centered around zero than the NO and BL experiments.



**Table 1.** GOCART aerosol optical properties used to calculate the extinction coefficient at 550 $\mu$m for dry air.

| Aerosol Species | Median Diameter ($\mu$m) | Effective Radius ($\mu$m) | Geometric Standard Deviation ($\mu$m) | Density ($g\ cm^{-3}$) | Hygroscopicity ($\kappa$) | Index of Refraction (Real) | Index of Refraction (Imag) | Mass Extinction Efficiency |
|---|---|---|---|---|---|---|---|---|
| Sulfate | 0.138 | 0.242 | 2.03 | 1.7 | 0.61 | 1.524 | 1.00E-07 | 3.673 |
| OC1 (hydrophobic) | 0.037 | 0.087 | 2.2 | 1.8 | 0.05 | 1.524 | 6.00E-03 | 2.473 |
| OC2(hydrophilic) | 0.037 | 0.087 | 2.2 | 1.8 | 0.2 | 1.524 | 6.00E-03 | 2.473 |
| BC1 (hydrophobic) | 0.022 | 0.036 | 2 | 1 | 0 | 1.738 | 0.44 | 8.990 |
| BC2(hydrophilic) | 0.022 | 0.036 | 2 | 1 | 0.15 | 1.738 | 0.44 | 8.990 |
| SeaSalt1 | 0.171 | 0.3 | 2.03 | 2.2 | 0.8 | 1.495 | 1.00E-08 | 2.548 |
| SeaSalt2 | 0.571 | 1 | 2.03 | 2.2 | 0.8 | 1.495 | 1.00E-08 | 0.889 |
| SeaSalt3 | 1.856 | 3.25 | 2.03 | 2.2 | 0.8 | 1.495 | 1.00E-08 | 0.227 |
| SeaSalt4 | 4.283 | 7.5 | 2.03 | 2.2 | 0.8 | 1.495 | 1.00E-08 | 0.096 |
| Dust1 | 0.301 | 0.5 | 2 | 2 | 0.05 | 1.5242063 | 0.00800000038 | 1.596 |
| Dust2 | 0.842 | 1.4 | 2 | 2.6 | 0.05 | 1.5242063 | 0.00800000038 | 0.507 |
| Dust3 | 1.444 | 2.4 | 2 | 2.6 | 0.05 | 1.5242063 | 0.00800000038 | 0.275 |
| Dust4 | 2.708 | 4.5 | 2 | 2.6 | 0.05 | 1.5242063 | 0.00800000038 | 0.140 |
| Dust5 | 4.814 | 8 | 2 | 2.6 | 0.05 | 1.5242063 | 0.00800000038 | 0.078 |





**Table 2.** Bias values for the inner domain for the moving average (MA), small bootstrap (BS), and large bootstrap (BL) bias correction schemes for each DA cycle with AOD observations.

|          | MA       | BS       | BL       |
|----------|----------|----------|----------|
| Cycle 2  | 0.134862 | 0.134799 | 0.134845 |
| Cycle 3  | 0.078208 | 0.023696 | 0.023688 |
| Cycle 6  | 0.144743 | 0.564639 | 0.564231 |
| Cycle 7  | 0.211556 | 0.568451 | 0.569715 |



**Table 3.** For the NO, MA, and BL experiments, several diagnostic metrics are presented including percent change of the cost function ($\%\Delta_{cost}$) and RMS ($\%\Delta_{RMS}$) errors from the background to the analysis, the skill, spread, spread over skill, and $\chi^2$ values.

| | $N_{obs}$ | $\%\Delta_{cost}$ | $\%\Delta_{RMS}$ | Skill | Spread | Spread over Skill | $\chi^2$ |
|---|---|---|---|---|---|---|---|
| | | | NO Bias Correction | | | | |
| Cycle 2 | 3340 | -4.581 | -5.660 | 0.08 | 0.13 | 1.49 | 1.11 |
| Cycle 3 | 3819 | -39.13 | -18.886 | 0.06 | 0.15 | 2.64 | 0.86 |
| Cycle 6 | 534 | -1.68 | -5.791 | 0.10 | 0.35 | 3.35 | 0.72 |
| Cycle 7 | 736 | 16.181 | -10.920 | 0.09 | 0.32 | 3.57 | 2.59 |
| | | | MA Bias Correction | | | | |
| Cycle 2 | 3243 | 4.753 | -6.936 | 0.10 | 0.13 | 1.32 | 1.23 |
| Cycle 3 | 3528 | 16.98 | -12.531 | 0.07 | 0.15 | 2.28 | 3.89 |
| Cycle 6 | 634 | -16.259 | -7.977 | 0.11 | 0.34 | 2.97 | 0.68 |
| Cycle 7 | 779 | -5.768 | -3.004 | 0.09 | 0.35 | 3.79 | 0.85 |
| | | | BL Bias Correction | | | | |
| Cycle 2 | 3243 | 4.695 | -6.700 | 0.10 | 0.13 | 1.33 | 1.23 |
| Cycle 3 | 3788 | -38.979 | -20.089 | 0.06 | 0.15 | 2.56 | 0.57 |
| Cycle 6 | 554 | -4.032 | -10.070 | 0.12 | 0.25 | 2.13 | 1.29 |
| Cycle 7 | 278 | 12.764 | -8.058 | 0.17 | 0.35 | 2.05 | 0.80 |



**Table 4.** Skewness and kurtosis for each bias scheme for DA cycles with AOD observations for the inner domain. Note the large values for these statistics for the BL experiment in the last cycle.

|         | NO Bias Correction | | MA Bias Correction | | BL Bias Correction | |
|---------|----------|----------|----------|----------|----------|----------|
|         | Skewness | Kurtosis | Skewness | Kurtosis | Skewness | Kurtosis |
| Cycle 2 | 0.4775   | 3.6186   | 0.8310   | 3.3329   | 0.8310   | 3.3329   |
| Cycle 3 | 0.5272   | 3.4957   | 0.7203   | 4.2091   | 0.4397   | 3.1446   |
| Cycle 6 | 0.1433   | 3.0499   | 0.5741   | 3.3338   | 0.1277   | 2.4534   |
| Cycle 7 | -0.0705  | 2.2001   | 0.3107   | 2.7962   | 2.7735   | 15.3375  |