# Peer review of "Bias correction in assimilation of AOD observations with WRF-Chem"

_Atmospheric Chemistry and Physics, 2018_

## Referee Comment (RC1) · Anonymous Referee #1 · 17 Dec 2018

This manuscript deals with BIAS correction of after the assimilation of observations of Aerosol Optical Depth into an ensemble system with WRF-Chem. This problem is not new, however I think that this work can become an important contribution since (a) the proper 3D representation of aerosols in the littoral area (or any area) is not an easy task and is one that is finally being tacked, (b) it helps in using existing satellite observations in more precise. Of course one would like to avoid these pesky biases from the beginning, but since then correcting them is what can be done.

I believe this article can be accepted after the following considerations (and of course those from other reviewers).

Major comments 1. I am not aware if there has been work on bias correction in the case of aerosols and satellite data. However, the bias correction problem has been

studied in the past. I am quite sure some operational centres (e.g. UK Met Office and ECMWF) which have included the presence of biases in their cost functions, and solve this problem in a direct or approximate manner. This should be mentioned and reviewed in the introduction. Probably some off-line methods are also used. 2. As with any study that involves a type of verification/validation, it is always better to have an independent set of observations to validate against, instead of validating with respect to the same observations used in the assimilation. Is this possible in your case? Can you at least cross-validate by partitioning the observations that you have and just using some for the assimilation and bias correction? If not, do you think this would change the results? 3. Some more details of the MA and Boostrap experiments are required. In the MA experiments, it is noted that the procedure can be done regionally. In this study cases I guess that this was done for the whole region of interest, is this correct? How would a 'regional' method look like? I guess one can partition by vertical levels, etc. I am not asking to do this (if it is too much trouble), but at least discuss it. For the bootstrap method, can you say anything about the changes in sample statistics (mean and covariance for instance) of the innovations (after the sampling with replacement). How does this change as the number of resampled elements increased. I am just thinking of the fact that you have only used two sizes differing by an order of magnitude. Is it because at those values convergence had been achieved?

Clarifications 1. Page 2, Line 12. The definition of bias is incomplete. From the text one could have the wrong impression that a bias is an instantaneous difference between the estimator and the true value. I think an expected value is missing. 2. Page 3, Line 25. The description of Pf in terms of the 'square' of a matrix of perturbations is not quite clear. Please re-formulate. 3. Page 5, Line 8. It was not clear to me if the h operator is an existing one (from previous works) or if it was developed and/or adapted for this work. 4. Section 3, line 15. Can you indicate why is necessary to show diagnostics both in the model and observation space? Are there differences on what can be measured in each space? 5. Sensitivity to ensemble size? If not, at least some description of the inflation and/or localisation that is used in the DA of your experiments. Does the

localisation differ from the original (meteorological) state variables of the model to the extended variables (which include the variables related to the aerosols).

Format 1. No indent is necessary when text continues after an equation (and of course it is not a new paragraph). Use \noindent. 2. Be consistent in the way you denote operations in the equations. Some times the scalar product (simple multiplication) is expressed with a cross (x), whereas in other occasions it is represented with a dot (.). 3. Some expressions would benefit from superindices (or subindices), such as in the case of departures from observations: y-h(xˆb), or y-h(xˆa) to indicate if they refer to background or analysis. 4. Equation 8. I do not think it is necessary to write the RMSE equation twice. 5. Section 3.1. I think the manuscript would have a more logical flow if this section were introduced after 2.6 when the diagnostics are mentioned. Then the result section would not need to be interrupted to discuss information theory. 6. Tables 2 and 3 are a great way to summarise the results. I wonder if it is worth to represent some of these results in a graphical manner.

---

## Referee Comment (RC2) · Anonymous Referee #2 · 10 Jan 2019

General comment: The primary objective of this study is to improve 3D aerosol structure over littoral zones. To improve the 3D aerosol structure, the authors assimilated MODIS AOD from Aqua and Terra Satellites using a coupled modeling system, consisting of a MLKF (Maximum Likelihood Ensemble Filter) analysis system and WRF-Chem. A case study that occurred over the Arabian Peninsula was used as a demonstration. The authors argued that systematic differences between observations and model guess (i.e. short-term forecast) exist, defined as biases. Thus the authors proposed to use two different bias correction methods, one "moving average (MA)" and the other "bootstrap", to correct AOD observations before assimilation. Three experiments were conducted: one without AOD bias correction and the other two with each of the bias correction methods. Metrics of diagnosed parameters were used to evaluate the

analyses among these three experiments. They concluded that the result with the assimilation of MA bias corrected AOD data was slightly better than that without any bias correction and that the result from the assimilation of bootstrap bias corrected data was the worst. The study is interesting and important. However, there are some major concerns that need to be addressed before this article can be accepted for publication in ACPD.

Major comments: 1. The introduction did not provide enough review on AOD data assimilation. It needs to be significantly improved. 2. The difference between observed AOD and model guess could be due to model bias too. While it sounds like a good idea to apply the bias correction to AOD observations before they are assimilated, it is possible that model biases can be aliased into observation biases, in particular when the bias is defined as the difference between observation and model guess (short-term forecast). The use of another independent observation to verify the analysis is important to this study. In addition, as the purpose of developing the coupled system is to reduce forecast uncertainty, it makes sense to extend the work to include some forecast results, which is another way to evaluate the use of the AOD data with and without bias correction. 3. Are there meteorological observations, either conventional or remotely sensed data, assimilated in these numerical experiments? To more accurately evaluate the impact of AOD assimilation on analysis, other meteorological observations have to be assimilated in order to do a fair comparison. In addition, were the aerosol-cloud-radiation interactions included in the model forecast (i.e. model guess)? The neglect of these physical processes can cause some model aerosol biases, which can then be aliased into observation biases. 4. When evaluating DA analysis, were observations used in evaluation also bias corrected for those experiments using bias corrected data? This will have a great impact on the evaluation results, but it is not clearly stated in the manuscript. If the answer is yes, then the use of another independent observation for verification is important in this study.

Minor comments: Page 2, line 29" "the controls variables are . . . " Page 3: what are the

radiation and boundary layer schemes that were used in model simulations? Page 4, line 1: "The control variables includes . . ." Page 5, line 5: Shouldn't Terra satellite pass at about 10:30 am local time, instead of 11:30 am? Page 5, line 11: Does GOCART take care of aerosol aging processes? If not, use "internally mixed" instead of "externally mixed". Page 7, line 21, what is the "control estimates" here? Is it the ensemble mean? Clarify it. Page 8. Line 21: the "maximization" of the cost function. Fig. 3 looks more like a table, instead of a figure. Fig. 4. Why was there no AOD data over eastern side of North Africa? Is there any quality issue of AOD data over there? Thirty-two ensemble members are used in the study. What is the localization value used in data assimilation? Is it the same for both domains?
* * *

---

## Author Comment (AC1) · 28 Feb 2019

Interactive comment on "Bias correction in assimilation of AOD observations with WRF-Chem"

Anonymous Referee #1

This manuscript deals with BIAS correction of after the assimilation of observations of Aerosol Optical Depth into an ensemble system with WRF-Chem.  This problem is not new, however I think that this work can become an important contribution since (a) the proper 3D representation of aerosols in the littoral area (or any area) is not an easy task and is one that is finally being tackled; (b) it helps in using existing satellite observations in more precise.  Of course one would like to avoid these pesky biases from the beginning, but since then correcting them is what can be done.

I believe this article can be accepted after the following considerations (and of course those from other reviewers).

*Author response:*  We would like to thank the reviewer for his/her valuable input.  We have addressed each of your comments and reference the appropriate changes in the manuscript.  Hopefully with these changes the reviewer would encourage a resubmission of this manuscript.

Major comments
1.  I am not aware if there has been work on bias correction in the case of aerosols and satellite data.  However, the bias correction problem has been studied in the past.   I am quite sure some operational centres (e.g. UK Met Office and ECMWF) which have included the presence of biases in their cost functions, and solve this problem in a direct or approximate manner.  This should be mentioned and reviewed in the introduction. Probably some off-line methods are also used.

*Author response:*  To the authors' knowledge bias correction for satellite aerosol assimilation has not been previously addressed.  But the reviewer is correct that operational centers do employ bias correction methods for radiance assimilation and this should be mentioned in the introduction.  The introduction will now include the following paragraphs:

"To the authors' knowledge bias correction has not yet been investigated for assimilating AOD observations.  However, this problem has been previously addressed for satellite radiance assimilation.  These bias correction procedures can be generally categorized as either static (offline) or variational.  The static bias correction scheme (Eyre (1992)) considers differences in the observations and the model state over a period of time and defines bias predictors using satellite scan angle along with several atmospheric variables (e.g. skin temperature, total column water, etc.).  This is carried out offline for each satellite sensor and band and is frequently updated.  The bias correction is then applied to the observations in the DA system.  The operational scheme was developed at the European Centre for Medium-Range Weather Forecasts (ECMWF) (Harris and Kelly (2001)) and is also employed at the Met Office (Hilton et al. (2009)).

Variational bias correction methods include bias coefficients within the state vector of the minimized cost function.  Therefore these coefficients are continuously updated, along with the state vector itself, during each DA cycle.  The bias is defined as a linear combination of predictors, similar to the static scheme, using scan angle along with atmospheric variables.  More details can be found in Derber et al. (1991), Parrish and Derber (1992); Derber and Wu (1998), Dee (2005), and Auligne et al. (2007)."

2. As with any study that involves a type of verification/validation, it is always better to have an independent set of observations to validate against, instead of validating with respect to the same observations used in the assimilation. Is this possible in your case? Can you at least cross-validate by partitioning the observations that you have and just using some for the assimilation and bias correction? If not, do you think this would change the results?

*Author response:* Yes, verification against independent observations was sought after. One option considered was the Aerosol Robotic Network (AERONET) but as there was only 1 or 2 sites available over the time period this proved to be inadequate as the DA system minimizes the cost function over the whole analysis domain. CALIOP data from the CALIPSO satellite was also considered but could not be used due to infrequent passage over this domain during our case study. This will be noted in the Results part (Section 3) of the manuscript. We are currently seeking data from the Navy Aerosol Analysis and Prediction System (NAAPS) reanalysis product to serve as an independent verification source but cannot currently report any results.

We did not consider partitioning our observations as we were more concerned with assimilating all of the observations that were available in the coastal region of our domain and viewing the response from the bias correction schemes. There simply were not enough observations to conduct an accurate statistical analysis if we removed a subset of observations and performed the experiments without these observations. If we had tried this technique it is hard for me to predict how much, if any, our experiments would have performed.

3. Some more details of the MA and Boostrap experiments are required. In the MA experiments, it is noted that the procedure can be done regionally. In this study cases I guess that this was done for the whole region of interest, is this correct? How would a 'regional' method look like? I guess one can partition by vertical levels, etc. I am not asking to do this (if it is too much trouble), but at least discuss it. For the bootstrap method, can you say anything about the changes in sample statistics (mean and covariance for instance) of the innovations (after the sampling with replacement). How does this change as the number of resampled elements increased. I am just thinking of the fact that you have only used two sizes differing by an order of magnitude. Is it because at those values convergence had been achieved?

*Author response:* The term "regional" was used for the MA scheme since it is applied to this specific domain in time and space. In the Conclusions section we state "The first bias correction scheme is described as a moving average that accumulates and incorporates the bias found in all previous assimilation cycles in an attempt to capture and correct for the regional and temporal representation error characteristics." This will be noted more clearly in the section (2.4.1) describing the MA scheme. We have not considered partitioning by vertical level but this may of interest in future research.

Regarding the bootstrap method, two experiments were conducted with different sample sizes – "small bootstrap" (M=10,000) and "large bootstrap" (M=100,000). The "small" $M$ was chosen for robust sample statistics (mean, variance). The "large" $M$ was chosen to determine if a much larger sample would significantly change the results and justify the extra computational cost. From comparing these experiments it became evident that the "small" $M$ gave incredibly similar results to the "large" $M$. This was true for seemingly all evaluated diagnostics and DA analyses. This was noted and therefore only results from the "large" $M$ experiment were presented. In the Results section we state "Table 2

contains the biases found for each scheme per assimilation cycle with AOD observations. These values indicate how the model guess is adjusted prior to the maximization of the cost function (Eq. 1). From this table it is evident that both the BL and BS schemes compute essentially the same bias. For this reason, and since multiple other diagnostics seem to indicate similar results between the BL and BS schemes, only results from the BL experiment will be presented." We will add to this paragraph a description of the "small" and "large" $M$ and the incredibly similar results from these experiments.

Clarifications
1. Page 2, Line 12. The definition of bias is incomplete. From the text one could have the wrong impression that a bias is an instantaneous difference between the estimator and the true value. I think an expected value is missing.

*Author response:* Yes, this clarification should be made. Page 2, line 20 has been changed to "Bias refers to the expected value of the difference between the true value and the one predicted by an estimator." Also, on page 4, line 24, equation (2) now defines bias to be $\mathbf{E}[\mathbf{y}-h(\mathbf{x})]$.

2. Page 3, Line 25. The description of Pf in terms of the 'square' of a matrix of perturbations is not quite clear. Please re-formulate.

*Author response:* For simplicity these few sentences have been eliminated. Page 4, line 11 now simply states "The matrix $\mathbf{P_f}$ is the forecast error covariance and the diagonal matrix $\mathbf{R}$ is the observation error covariance." Appropriate references for the MLEF are present in the manuscript which includes a more thorough description of the DA system. More formal details of the MLEF DA system can be included in the revised manuscript if desired.

3. Page 5, Line 8. It was not clear to me if the h operator is an existing one (from previous works) or if it was developed and/or adapted for this work.

*Author response:* Yes, this is an existing operator from previous research (Pagowski et al. 2014). This sentence has been corrected with the appropriate citation.

4. Section 3, line 15. Can you indicate why is necessary to show diagnostics both in the model and observation space? Are there differences on what can be measured in each space?

*Author response:* Both model- and observational-space are presented because it is these two components that comprise the DA cost function. While both pieces are not typically presented in the results section of a DA paper, we thought it interesting to present both here. While the analysis is most often shown, the observational space results also show the number and location of the assimilated observations. This is now noted in the beginning of section 3.

5. Sensitivity to ensemble size? If not, at least some description of the inflation and/or localisation that is used in the DA of your experiments. Does the localisation differ from the original (meteorological)

state variables of the model to the extended variables (which include the variables related to the aerosols)?

*Author response:* These experiments have only been run with 32 ensemble members that utilize both covariance inflation and covariance localization. For localization the method described by Yang et al. (2009) is used with horizontal correlation length scales of 250 km and 100 km in the outer and inner domains respectively and a vertical length scale of 0.3 sigma levels. We keep these constant throughout the experiments for all state variables (meteorological and aerosols). This will be noted in Section 2.2.

Format
1. No indent is necessary when text continues after an equation (and of course it is not a new paragraph). Use \noindent.

*Author response:* We appreciate the reviewer pointing this out and all instances have been corrected.

2. Be consistent in the way you denote operations in the equations. Sometimes the scalar product (simple multiplication) is expressed with a cross (x), whereas in other occasions it is represented with a dot (.).

*Author response:* We appreciate the reviewer pointing this out and all instances have been corrected.

3. Some expressions would benefit from superindices (or subindices), such as in the case of departures from observations: y-h(xˆb), or y-h(xˆa) to indicate if they refer to background or analysis.

*Author response:* We have checked all of these equations and have made clarifications when possible.

4. Equation 8. I do not think it is necessary to write the RMSE equation twice.

*Author response:* We have combined these two equations and included subindices to define them.

5. Section 3.1. I think the manuscript would have a more logical flow if this section were introduced after 2.6 when the diagnostics are mentioned. Then the result section would not need to be interrupted to discuss information theory.

*Author response:* We agree and the description of information theory has been moved to the end of Section 2.

6. Tables 2 and 3 are a great way to summarise the results. I wonder if it is worth to represent some of these results in a graphical manner.

*Author response:* We had considered presenting this data in a graph but were hesitant due to the fact it is not "continuous" in that observations were assimilated in cycles 2, 3, 6, and 7 (not 1, 4, and 5). If the data were plotted we were concerned a reader may interpret that these diagnostics pertain to cycles 1-7. We also considered a bar chart, with each diagnostic listed above the cycle number, but it appeared

quite jumbled and difficult to focus on the individual values due to the different scales of the data.  We are certainly open to other suggestions on how to present this data that may be beneficial to the reader.

Anonymous Referee #2

General comment: The primary objective of this study is to improve 3D aerosol structure over littoral zones. To improve the 3D aerosol structure, the authors assimilated MODIS AOD from Aqua and Terra Satellites using a coupled modeling system, consisting of a MLKF (Maximum Likelihood Ensemble Filter) analysis system and WRF-Chem. A case study that occurred over the Arabian Peninsula was used as a demonstration. The authors argued that systematic differences between observations and model guess (i.e. short-term forecast) exist, defined as biases. Thus the authors proposed to use two different bias correction methods, one "moving average (MA)" and the other "bootstrap", to correct AOD observations before assimilation. Three experiments were conducted: one without AOD bias correction and the other two with each of the bias correction methods. Metrics of diagnosed parameters were used to evaluate the analyses among these three experiments. They concluded that the result with the assimilation of MA bias corrected AOD data was slightly better than that without any bias correction and that the result from the assimilation of bootstrap bias corrected data was the worst. The study is interesting and important. However, there are some major concerns that need to be addressed before this article can be accepted for publication in ACPD.

*Author response:* We would like to thank the reviewer for his/her valuable input. We have addressed each of your comments and reference the appropriate changes in the manuscript. Hopefully with these changes the reviewer would encourage a resubmission of this manuscript.

Major comments:
1. The introduction did not provide enough review on AOD data assimilation. It needs to be significantly improved.

*Author response:* We agree that AOD DA should be thoroughly reviewed in the introduction. The following paragraph will now be included:

"Recent research involving the assimilation of chemistry and aerosol data (Collins et al., 2001; Wang et al., 2003; Weaver et al., 2007; Wang and Niu, 2013; Zhang et al., 2014, Randles et al., 2017) has been applied to both global (Zhang et al. 2008b; Uno et al. 2008; Benedetti et al. 2009) and regional (Pagowski et al. 2010; Liu et al. 2011) models. Column-integrated aerosol data assimilation schemes (Weaver et al. 2007; Zhang et al. 2008b) have also proven to be efficient as well. Ensemble Kalman Filter (EnKF) data assimilation methods have been employed (Pagowski and Grell 2012; Rubin et al., 2016, 2017) as well which includes a flow-dependent error covariance matrix to more accurately model the forecast error covariance for aerosol/chemistry variables. Hybrid aerosol data assimilation applications (Schwartz et al. 2014; Pagowski et al. 2014) have also offered improved representation of forecast error covariance and the ability to address nonlinear interactions."

2. The difference between observed AOD and model guess could be due to model bias too. While it sounds like a good idea to apply the bias correction to AOD observations before they are assimilated, it is possible that model biases can be aliased into observation biases, in particular when the bias is defined as the difference between observation and model guess (short-term forecast). The use of another independent observation to verify the analysis is important to this study. In addition, as the purpose of developing the coupled system is to reduce forecast uncertainty, it makes sense to extend the work to include some forecast results, which is another way to evaluate the use of the AOD data with and without bias correction.

*Author response:* Yes, we agree that model bias could certainly be a factor and using another set of independent observations as verification would prove to be beneficial. One option considered was the Aerosol Robotic Network (AERONET) but as there was only 1 or 2 sites available over the time period this proved to be inadequate as the DA system minimizes the cost function over the whole analysis domain. CALIOP data from the CALIPSO satellite was also considered but could not be used due to infrequent passage over this domain during our case study. This will be noted in the Results part (Section 3) of the manuscript. We are currently seeking data from the Navy Aerosol Analysis and Prediction System (NAAPS) reanalysis product to serve as an independent verification source but cannot currently report any results.

As the focus of this manuscript was on improving the analysis via different bias correction techniques we believe considering forecast results outside of the scope of this research. We would also encounter the same problem of not having ample data for verification of the forecasts. In the future we could investigate forecast results but this current research goal was inspecting the analysis from a data assimilation system composed of different de-biasing methods. We hope the reviewer can appreciate these preliminary steps toward improving forecasting by first considering the impact on the analysis from assimilating AOD observations.

3. Are there meteorological observations, either conventional or remotely sensed data, assimilated in these numerical experiments? To more accurately evaluate the impact of AOD assimilation on analysis, other meteorological observations have to be assimilated in order to do a fair comparison. In addition, were the aerosol-cloud-radiation interactions included in the model forecast (i.e. model guess)? The neglect of these physical processes can cause some model aerosol biases, which can then be aliased into observation biases.

*Author response:* In these experiments there are no meteorological observations assimilated. The only assimilated observations are that of AOD. This will be made clear in Section 2.3.1. The primary concern of this paper is to evaluate bias correction methods for AOD observations and we believe assimilating meteorological observations is out of the scope of this research. In fact, there are two other manuscripts currently under review (Wu et al., 2019; Zupanski et al., 2019) for this ACP/AMT special issue (Holistic Analysis of Aerosol in Littoral Environments - A Multidisciplinary University Research Initiative) that assimilates meteorological observations along with AOD for this particular case study. We believe that as those results are quite substantial they should remain in a separate paper while this manuscript focuses solely on bias correction methods. The model forecast did not include any aerosol-cloud-radiation interactions. We understand that this could cause some model aerosol biases and this will be noted in Section 2.1.

4. When evaluating DA analysis, were observations used in evaluation also bias corrected for those experiments using bias corrected data? This will have a great impact on the evaluation results, but it is not clearly stated in the manuscript. If the answer is yes, then the use of another independent observation for verification is important in this study.

*Author response:* The presented results include discussion of the analysis increments ($\mathbf{x}\_a - \mathbf{x}\_b$) and the innovations (y-h($\mathbf{x}$)). We do not use the assimilated observations to verify our analyses. As stated above, we sought independent observations for verification but so far have been unsuccessful.

Minor comments:

Page 2, line 29" "the controls variables are..."

*Author response:*  This correction has been made.

Page 3: what are the radiation and boundary layer schemes that were used in model simulations?

*Author response:* The New Goddard longwave and shortwave radiation scheme was used along with the YSU boundary layer scheme.    This is now included in section 2.1.

Page 4, line 1: "The control variables includes..."

*Author response:*  This correction has been made.

Page 5, line 5: Shouldn't Terra satellite pass at about 10:30 am local time, instead of 11:30 am?

*Author response:*  Yes, you are correct and this has been changed in the manuscript.

Page 5, line 11:  Does GOCART take care of aerosol aging processes?  If not, use "internally mixed" instead of "externally mixed".

*Author response:*  No, GOCART does not take aerosol aging into account.  GOCART tracks the aerosol mass externally as there are discrete output variables for dust, sea salt, and sulfate.  So it is externally mixed.

Page 7, line 21, what is the "control estimates" here?  Is it the ensemble mean? Clarify it.

*Author response:*  The control estimates refer to the forecast produced (from the model) after the DA analysis.  Spread is then calculated from the ensemble and this forecast.  This will be noted in the revised manuscript.

Page 8. Line 21: the "maximization" of the cost function.

*Author response:*  This has been changed to "minimization."

Fig. 3 looks more like a table, instead of a figure.

*Author response:*  This can be re-labeled as a table.

Fig. 4. Why was there no AOD data over eastern side of North Africa?  Is there any quality issue of AOD data over there?

*Author response:*  The MODIS 3-km AOD products over land are retrieved through the Dark-target algorithm, which is only suitable for dark-surfaces (vegetation and soils).   Thus there are no AOD retrievals over most of North Africa and the Middle East where the land is predominately desert or arid bare surface.  This will be noted in Section 2.4.

Thirty-two ensemble members are used in the study.  What is the localization value used in data assimilation? Is it the same for both domains?

*Author response:*  These experiments have only been run with 32 ensemble members that utilize both covariance inflation and covariance localization.  For localization the method described by Yang et al. (2009) is used with horizontal correlation length scales of 250 km and 100 km in the outer and inner domains respectively and a vertical length scale of 0.3 sigma levels. We keep these constant throughout the experiments for all state variables (meteorological and aerosols).  This will be noted in Section 2.2.